# A Qualitative and Comprehensive Analysis of Caries Susceptibility for Dental Fluorosis Patients

**DOI:** 10.3390/antibiotics10091047

**Published:** 2021-08-27

**Authors:** Qianrui Li, Jiaqi Shen, Tao Qin, Ge Zhou, Yifeng Li, Zhu Chen, Mingyun Li

**Affiliations:** 1State Key Laboratory of Oral Diseases, National Clinical Research Center for Oral Diseases, West China School of Stomatology, Sichuan University, Chengdu 610041, China; 2018151642180@stu.scu.edu.cn (Q.L.); 2018151642094@stu.scu.edu.cn (J.S.); 2018151642095@stu.scu.edu.cn (T.Q.); 2018151642170@stu.scu.edu.cn (G.Z.); 2018151642169@stu.scu.edu.cn (Y.L.); 2Key Laboratory of Oral Disease Research, School of Stomatology, Zunyi Medical University, Zunyi 563000, China; m17784810646@163.com

**Keywords:** dental fluorosis, caries susceptibility, fluoride, enamel defects, salivary buffer capacity, eating habits

## Abstract

Dental fluorosis (DF) is an endemic disease caused by excessive fluoride exposure during childhood. Previous studies mainly focused on the acid resistance of fluorotic enamel and failed to reach a consensus on the topic of the caries susceptibility of DF patients. In this review, we discuss the role of DF classification in assessing this susceptibility and follow the “four factors theory” in weighing the pros and cons of DF classification in terms of host factor (dental enamel and saliva), food factor, bacteria factor, and DF treatment factor. From our analysis, we find that susceptibility is possibly determined by various factors such as the extent of structural and chemical changes in fluorotic enamel, eating habits, fluoride levels in diets and in the oral cavity, changes in quantity and quality of saliva, and/or oral hygiene. Thus, a universal conclusion regarding caries susceptibility might not exist, instead depending on each individual’s situation.

## 1. Introduction

Dental fluorosis (DF) is an enamel malformation caused by the chronic intake of excessive fluoride during tooth development, and is characterized by hypomineralization (or porosity). This porous characteristic is demonstrated by clinical features ranging from enamel opacities, discoloration, or stains to structural defects (pits, cracks, and fissures) [1,2]. However, despite its intimate relationship with fluoride, which is widely recognized as “the cornerstone of dental caries reduction” [3], the enamel caries susceptibility of DF has remained ambiguous [4]. Epidemiological surveys have demonstrated contradictory conclusions. While some authors suggest that there is a conspicuous negative association between fluorosis status and caries experience [5], others observed that 0–5-year-old children affected by DF were more likely to develop early childhood caries [6]. All these opposite conclusions indicate that a universal conclusion explaining caries susceptibility in this case might not exist, as in each case some factors were disregarded. Thus, a comprehensive analysis of dental caries susceptibility for this population group is needed to inform decisions regarding the prevention of caries for DF patients. To address the question of whether DF patients are more or less prone to dental caries, and using the guideline of the “four factors theory”, we attempt to explore the morphological and compositional changes of fluorotic enamel; the quantity and quality changes in patients’ saliva; the features of food in endemic areas; the interaction between oral bacteria and the host; possible treatment procedures for DF that might incur caries and assess effects of these factors on the caries susceptibility. Given that: a. DF bears an “endemic nature” [7,8], namely happening mostly in high fluoride areas; b. the DF prevalence in deciduous dentition is relatively low [9,10], which might owe to the function of the placental barrier, regulating the infusion of F from maternal to fetal blood, and the poor transportation of F from plasma to breast milk [11]; and c. dental enamel is at the frontier, where acid produced by bacteria first attacks and signs of caries tend to earliest appear, to conduct our analysis we mainly focus on the corresponding situation: permanent teeth affected by DF in endemic areas. Original studies published in August 2021 were chosen through literature searches in Web of Science, PubMed and EMBASE database. We searched for studies that discussed the relationship between DF and dental caries, the morphological and compositional changes of DF in the oral cavity, the influence of mastication exertion on saliva secretion, and the environmental features of high fluoride areas. Of note, we only accepted studies written in English and mainly focused on humans aged more than 12 years. However, restrictions on age or even species were compromised if there was a lack of pertinent literature. We used both controlled vocabulary and free text terms and inspected the citations and references of relevant studies.

## 2. Dental Fluorosis Classification

DF classification is of significant importance for caries susceptibility analysis as patients at different stages may manifest different predispositions to caries. Traditionally, the most well-received classification systems are those based on the clinical severity (appearance) of fluorotic teeth, i.e., clinical indices such as Dean’s Index and the Thylstrup Fejerskov (TF) Index [12]. It was detected that increasing severity parallels the increasing porosity of the subsurface enamel all along the tooth [13]. When faced with a cariogenic challenge, it is the subsurface apatite crystals that are dissolved, consequently causing a subsurface lesion [14]. Accordingly, it can be reasonably concluded that a fluorotic tooth displaying higher clinical severity represents a wider mineral area to be dissolved by acids and the permeability of acid through enamel might be elevated [4]. In more clinically severe forms of DF post-eruptive damage happens, revealing less caries-resistant subsurface (see descriptions below). Similarly, attrition causing the removal of surface enamel lesions can mitigate symptoms, especially in the milder form cases. A recent longitudinal study confirmed dynamic post-eruptive changes with teeth presented reduced, increased and the same scores after a 3-year follow-up [15]. Therefore, it is evident that tremendous potential concerning the evaluation of caries susceptibility lurks in the clinical classification system.

However, these indices are basically subjective, and examiner bias may possibly exist [16]. Today, techniques such as fluorescence imaging and deep learning may appear as objective and promising means to quantify DF [12,17]. Another main source of misinterpretation comes from post-eruptive changes: attrition causing the removal of surface enamel can mitigate symptoms especially in the milder form cases; the maxillary incisors are prone to air exposure, particularly the incisal part, and to drying out, resulting in a clearer view of enamel surface porosities; and the question as to whether to comprise enamel stain in the scoring criteria, since personal dietary habits exert influence on the uptake of stain [1].

## 3. Host Factor

### 3.1. Dental Enamel

Fluorotic enamel generally contains more fluoride [18], which substitutes the OH^-^ in hydroxyapatite to form a more acid-resistant mineral: fluorapatite. Here we define the “critical pH” with respect to a certain mineral as the pH at which dissolution equilibrium status occurs. The critical pH for hydroxyapatite and fluorapatite (5.5 & 4.5, respectively) apparently demonstrates that the solubility of hydroxyapatite increases 10 times with regard to fluorapatite [14].

In spite of the higher fluoride concentration, no consensus on the issue of the relationship between enamel fluorosis severity and fluoride content has been reached [19]. Some research support a positive correlation between the two in both human and rat [18,20,21,22], with each fluorosis severity category bearing a large standard deviation and overlapping fluoride concentration. While others claim that enamel surface features are irrelevant to enamel fluoride concentration, as the dentin fluoride concentration displays a correlation with DF severity [23]. The lack of consensus, aside from systematic errors in experimental methods, can be partly attributed to the interference of genetic factors. Researchers have noted that murine experiments presented the fact that different strains of mice bearing the same amount of F in mineralized tissue could yield variable degrees of DF severity [24]. As they suggested, there is growing evidence that hereditary background could exert major influence on inter-individual variation in predisposition to fluorosis. 

Some researchers employed unerupted fluorotic teeth, which fail to undergo post-eruptive remineralization, to conduct their research. Since fluorapatite is less soluble in acid, the presence of fluoride will facilitate the enamel remineralization process. In the aforementioned description, during a cariogenic challenge, hydroxyapatite crystals are dissolved from the subsurface. However, if there is the presence of fluoride then a layer rich in fluorapatite crystals is formed at the surface, resulting in a well-mineralized “surface layer”. The fluoride profile showed that the highest fluoride concentrations of fluorotic enamel steadily appeared at the outermost, beneath which a drastic fall in fluoride concentrations was observed, confirming the existence of such “surface layer” [25]. In the light of the porosity of fluorotic enamel, it may be plausible to infer that the post-eruptive uptake of fluoride will happen to a more remarkable degree and thus we might be able to partly attribute the substantial discrepancy in the same erupted group to the greater amount of fluoride uptake [1]. On the other hand, while it has been observed that the formation of a well-mineralized surface zone formation in fluorotic enamel precedes the eruption [22], the contribution of post-eruptive fluoride uptake was not denied. A study aiming at this post-eruptive issue showed, through an in vitro experiment, that fluoride is not readily incorporated into sound enamel crystals after eruption [26]. The calcium-fluoride-like material may form on the outermost layers of enamel (including biofilm, pellicle, and sites in which porosity prevails) when higher levels of fluoride are presented both in vitro and in vivo, sheltering crystals from dissolution and acting as a pH-modulated reservoir to adsorb or release fluoride and calcium [14,27]. 

Over recent years, the prominent cariostatic mechanism of fluoride has been more related to its post-eruptive, topical effect (the calcium-fluoride-like material is a source of its topical effect), with focus shifting away from incorporated F, namely fluorapatite [28,29]. As has been mentioned in a review, no research was able to substantiate a lower caries prevalence in normal enamel with a high F content compared with a low F content condition [30]. Since fluoride concentration in enamel as high as in shark teeth, which is constituted by nearly pure fluorapatite crystal (about 30,000 ppm F), confers meager protection against caries, protection that is dwarfed by the regimen of a daily mouth rinse with 0.2% NaF [31]. Hence, it can be seemingly and reasonably extrapolated that the structurally incorporated fluoride concentrations in fluorotic enamel, even at the highest level, the surface layer, lack efficacy in containing caries progression.

Notwithstanding all the listed descriptions that accentuate the ineffectiveness of bound F in the face of cariogenic challenge, these studies have mainly pertained to the intact enamel cases which, in fluorotic teeth, might easily be obscured by other DF properties (see descriptions below). As for fluorotic enamel, such a verdict might not do justice to the F inside, although investigators have discovered that there were no differences with respect to lesion severity between fluorotic teeth and fluorotic teeth after formation of artificial caries lesions [32]. However, in adopting the abrasion procedure to standardize all teeth, this study did not preserve the enamel surface layer which might play a more critical role in fluorotic teeth than non-fluorotic teeth, taking their special physical structures into account. As mentioned above, DF involves hypomineralized (or porous) areas situated deeply within a well-mineralized surface layer. With increasing clinical signs, the porous transformation reaches deeper into the enamel and the extent of the porosity of the tissue also increases [1]. This abrasion procedure largely resembles the above-mentioned post-eruptive changes, especially in more severe cases. Compared with its non-fluorotic counterpart, fluorotic enamel displays a lower hardness and elastic modulus, and has an accordingly higher wear rate [33]. This positively indicates the diminished physical strength of fluorotic enamel, and that fluorotic enamel is more prone to fall victim to mastication forces in accordance with studies [34] that state that enamel thickness decreases with increased DF severity. The breakdown of the surface layer during cariogenic challenge leads to the exposure of more porous subsurface enamel and a more thorough diffusion of acid into the tooth. Interestingly, a cross-sectional study even saw a significant negative correlation between DF and dental caries at the tooth level in a population of schoolchildren. However, no relationship was detected at the individual level, affirming the role of bound F and showing the inability of topical effect fluoride [35]. As a consequence, the formation of this hypermineralized surface layer is protective, buffering the diffusion rate of demineralizing ions into the underlying fragile areas [36]. From another perspective however, it can be easily inferred that the presence of such a structure may render the remineralization of the lesion body more laborious when stuck in a dental caries condition.

Aside from the more porous surface change that post-eruptive damage delivers, extensive surface enamel loss arises beyond a certain level of porosity, rendering unevenness and forming pits on the surface [37,38]. In agreement with their conclusion, Hu D et al. [39] detected pores, pits, and other small defects on moderate to severe fluorotic enamel surface in contrast with the lack of obvious defects spotted on a sound enamel surface. It was noticed that, under a high degree of magnification, the hypermineralized surface formation comprised large hexagonal enamel crystals which were divided by spacious intercrystalline areas [40], hence the pores came into being, and plentiful irregular tiny crystals were noticed among these spacious areas. In places highly predisposed to caries, e.g., pits and fissures, however, it is the pre-eruptive fluoride that plays a predominant role in caries control [41], possibly because of the difficulty of access for the topical source of fluoride to these places. Regarding increasing structural defects on fluorotic surface enamel, increasing fluoride content in fluorotic enamel may even defend against pit and fissure caries more effectively. 

According to chemical analysis, in rodent fluorotic enamel specimens organic matter appeared to increase in the prism core and interprismatic regions, accompanied by lower crystallite density [42]. Wright et al. [43] described how protein content in permanent, moderately fluorosed enamel for subjects residing in endemic areas with 3.2 ppm F in drinking water, increased from 0.1% to 0.27% compared with non-fluorosed enamel. In human fluorotic lesions, enamel rods were demonstrated to be besieged by sheath-shaped organic substances [44], corresponding with the etiological feature of DF: the impaired removal of organic substances from enamel layers. Moreover, broken lesions in hypomineralized enamel of molar/incisor hypomineralization (MIH) were found to be able to incorporate proteins from saliva and blood over time, owing to their porous trait [45]. Considering that fluorotic enamel has the same porous property and exhibits subsurface porosity as MIH enamel does, exposure of the fluorotic enamel subsurface might also accumulate proteins. However, to verify this hypothesis, further research is required. As the organic substances pack the periprismatic areas, permitting the passage of small molecules and ion in and out of the tooth, a rise in the amount of the substances may promote enamel permeability. Meanwhile, from the perspective of specific inorganic components in enamel, magnesium concentrations increased while those of carbonate changed in the opposite direction. An increase in fluoride, which accords with the DF case, is usually paralleled by a rise in magnesium content, attributing to the high affinity between magnesium and fluoride [46]. In vitro studies simulating both hydroxyapatite formation and rodent animal experiments discovered the existence of magnesium-substituted amorphous calcium phosphate [47]. The substitution of magnesium may increase enamel solubility [48], which might be explained by its incompatibility in the crystal lattice, where it disturbs the ordered array of ions in the crystal [46]. Likewise, carbonate acts similarly to magnesium [49]. As mentioned before, the presence of fluoride accelerates the precipitation rate and dissolves carbonate ions easily [50], yielding a more structurally stable type of crystal (Figure 1).

### 3.2. Saliva

It may be safely said that saliva samples collected from DF patients in endemic areas contain higher levels of fluoride (see descriptions below), although the universal utility of fluoridated dentifrice might mask the significance of the difference between DF patients and healthy people. A clinical study conducted after the popularization of fluoridated dentifrice reported hardly any differences in salivary fluoride levels among primary-school-aged children both in the fluoridated and non-fluoridated community’s [51] compared with an earlier study undertaken prior to the popularity of fluoridated dentifrice had been reached, the difference was noticed [52]. It has been found that statistical analysis on the average level of decayed surfaces between an optimal water-fluoridated region and a low water-fluoridated region did not show significant differences, while more static lesions and fewer fillings were observed in the optimal water-fluoridated group, suggesting that the fluoride presented in the oral cavity mainly exercises its influence by promoting lesions arrest and regression (remineralization) rather than preventing the initiation of new lesions (demineralization) [53]. In cases where the outer layer is worn out, artificial lesions created on surface-abraded fluorotic enamel are more inclined to remineralization under fluoridated conditions than lesions created in sound teeth and display a greater amount of mineral precipitation. Greater porosity exhibited in fluorotic subsurface enamel provides more binding sites for fluoride [32], just as, in the same manner, subsurface enamel is more prone to demineralization (see descriptions above). Thus, it can be seen from this that the possibly elevated levels of salivary fluoride and the porous property of fluorotic teeth allow remineralization to happen to a more measurable degree. Another reported effect of fluoride is that the usage of fluoride-containing drinking water among rats brought about an increase in saliva flow rate [54]. A study found that the concentration of an important salivary component, sialic acid, which is capable of accelerating the aggregation of bacteria that contributes to the acquired pellicle formation and dental plaque, was reduced with increasing fluorosis severity [55]. Consistent with this finding, a study discovered that higher levels of secretory immunoglobulin A, an important antibacterial substance, in saliva samples from children with DF was negatively correlated with a decrease of sialic acid, though the difference was not significant [56].

Saliva produced by the parotid gland contains the highest level of bicarbonate [57], which is one of the most salient substances contributing to the buffer capacity of saliva. Especially under the mechanically stimulated state (chewing), bicarbonate dominantly accounts for the buffer capacity [58], coinciding with an increase in parotid gland secretion (constituting at least half of the whole mouth saliva, compared with 28% in the unstimulated condition) [59] due to stimulation of intra-oral mechanoreceptors and the initiation of the masticatory-salivary reflex [60]. The extent to which the secretion increases is determined by the applied stimulus intensity. Mastication is manipulated by the central pattern generator receiving plentiful sensory inputs as food is ingested and chewed in the oral environment (Figure 2) [61]. As previously described, fluorotic enamel features a reduced hardness, elastic modulus, and ability to resist mechanical wear, typical of severe forms of DF. This physically compromised enamel might impair the biting force exerted during mastication, bringing about changes such as extended chewing time, more or less frequent action of swallowing to compensate for the malfunction of teeth, and possibly altering the stimulus intensity. It has been observed that in ruminant animals, calves and buffaloes plagued by dental lesions (mottling, brownish stains, and deformity of the teeth) suggestive of fluorosis, appeared to suffer from painful mastication and have difficulty in mastication [62], and deer demonstrated diminished foraging efficiency resulting in no fat reserves [63], something that is highly in accordance with our hypothesis. In extreme cases DF patients may also suffer from malnutrition, which is associated with dental caries [64].

From the psychosocial dimension, some agreement has been reached in pointing out the view that cases with three or more TF scores may be recognized as an esthetic problem [65], especially severely affected anterior teeth [62], as defects and stains appears on the labial surface. The discoloration and deformity of DF patients’ teeth might cast an influence on the self-perception of their mastication and change their chewing habits, so that they refrain from opening their mouth wide and adopting a more elegant manner, for fear that others might spot the imperfections of their affected teeth, especially in public areas. 

According to the previous findings: the parotid gland produces saliva that is richest in bicarbonate and during mastication the parotid gland yields the largest proportion of saliva; with bicarbonate bearing the main responsibility for buffer capacity it might therefore stand to reason that in this period, the level of bicarbonate in saliva, the pH of saliva, and its buffer capacity all largely rely on the salivary flow rate, as their increases are guaranteed by an increase in the salivary flow rate, a deduction that has been confirmed [58]. The rate of salivary clearance, which refers to the dilution and elimination process of carbohydrates, acids, and bacteria [66], is regulated by the salivary flow rate, the pre- and post-swallowing volume of saliva presented in the oral cavity and the swallowing frequency [67]. Thus, we infer from the current studies that the potential changes of mastication pattern in DF patients might alter the volume and substance of saliva, the salivary buffer capacity, and the oral clearance effect. However, a study has suggested that the tendency for an increased flow rate with increasing DF severity was weak and was not statistically significant, with no variation in pH and buffer capacity between DF patients and healthy people [55].

Judging from a dynamic perspective, the aging process may influence the quantity and quality of saliva in DF patients. Unfortunately, no studies concerning this population have been published. Similar studies targeted at healthy people revealed substantial divergences of opinion: although an increased proportion of adipose and fibrovascular tissue was discovered in the salivary glands of elderly people [68], and much research has concluded that salivary flow rate decreases with age [69,70], no agreement has been reached on the change of quantity [71]; different studies have drawn opposite conclusions on the calcium concentration in the unstimulated saliva of elderly people [72,73]. Calcium and phosphate contribute to the maintenance of saliva in a supersaturation status with respect to all insoluble calcium phosphate salts and provide a reparative environment for keeping the enamel intact [74,75]. Aside from disagreement, it was noticed that the activity of lactoferrin and peroxidase (proteins that exhibit antibacterial property) was reduced in the same kind of population [76]. Particularly, the view that mucin concentration decreases with aging was widely accepted [71]. Mucin can interact with several strains of *Streptococcus mutans* and promote their agglutination, thus fostering the clearance of cariogenic bacteria from the oral environment [77]. On the other hand, mucin, evincing a high affinity to the hydroxyapatite, is a major component of acquired pellicle, which impairs the remineralizing ionic transport [78] and influences the adhesion of particular bacteria to the enamel surface. Moreover, a decreased mucin concentration might incur the loss of lubricating properties of saliva, bringing about poor wettability of enamel surfaces [71,79]. Theoretically, the protective role of these components is well supported, however, there is modest evidence for associations between experience of caries and a large variety of salivary parameters including all the items mentioned above [80,81]. The DF and healthy population might share some similarities with these salivary changes. Considering the mechanical properties of fluorotic enamel and some changes in the oral cavity during aging: exacerbating periodontal condition, missing teeth, and severely debilitated masticatory ability, salivary changes for DF patients may appear more enormous. Further research is required to disclose such changes.

## 4. Food Factor

There are three types of endemic fluorosis, including drinking water, brick tea and coal-burning fluorosis. For the drinking water type, the cause is groundwater contaminated by the dissolution of fluorine-bearing minerals is in excess of WHO recommended fluorine concentration standards [82]. For brick tea type, diet (food and beverage) in these endemic areas is highly reliant on brick tea, which is made of the older leaves of tea plants, containing the highest level of fluoride in the fluorine-rich plants [83]. For the coal-burning type, the combustion of high-fluorine coal and especially the binder clay on insufficiently ventilated stoves volatilize the majority of F, appearing in the form of smoke dust, part of which adheres to the surface of food while it is made [84,85]. Several studies have revealed that when people ingest meals prepared with fluoridated water or fluoridated table salt, saliva sampled during mastication and after meal ingestion exhibits significantly higher fluoride concentration [86,87]. This concentration recurs to the baseline level within half an hour to two hours, which might be explained by different amount of fluoride entered the oral cavity. A small portion of fluoride redistributed via the plasma to the saliva may also play a minor part in maintaining high-fluoride concentration and duration, as fluoride is absorbed through the gastrointestinal tract [88]. Considering the fluoride intake in diets of endemic areas is definitely more abundant than the systemic use of fluoride, and because the exposure frequency is guaranteed in diets, the elevated fluoride levels in the oral cavity of these residents might be sustained for a longer period. However endemic areas do not always assure a higher dietary fluoride intake, as effort was seen from some younger people in coal-burning type endemic areas in avoiding eating roasted corn and in replacing the staple food with rice, in the pursuit of a higher quality of life [89]. In addition, post-eruptive damage, as explained above, might also have the potential to change patients’ dietary habits, making them refrain from hard-textured, chewy ingredients, preferring instead soft foods that are more adhesive to the tooth surface, leading to extended food debris retention time. Considering salivary flow rate is highly responsible for the clearance of fluoride [90], the possible phenomenon of salivary secretion changes brought about by post-eruptive damage will complicate the fluoride level changes in oral cavity.

## 5. Bacteria Factor

Fluorotic teeth exhibiting rough surfaces with irregularities, which may bring about dental plaque retention [50] and foster adhesion of *Streptococcus mutans* [39,91], is generally regarded as the most salient element in caries formation. The morphological abnormalities of teeth surface might emerge as a major hinderance in maintaining oral hygiene, as research [92] has found that, in schoolchildren, oral cleaning was the most prominent daily routine burdened by DF. According to the analysis of saliva specimens collected from moderate to severe DF patients, two acidogenic bacterial species: *Streptococcus mitis* and *Lactococcus lactis* are found to dominate the oral microbiota, suggesting the active status of glycolysis presented in the oral cavity, and the poor oral condition was assumed to incur the shift of the oral microbiome [93].

From the microbiological aspect, fluoride presented in the oral cavity can retard bacterial growth and metabolization by inhibiting enolase and ATPase [94]. The degree of suppression should be consistent with the higher oral concentration of fluoride. In response to the fluoride challenge, bacteria are able to acquire resistance to fluoride. Although there are reports on the isolated fluoride-resistant strain found in endemic environments [95], the condition of the naturally arising, oral fluoride-resistant bacteria is scarcely known. Furthermore, fluoride levels in endemic areas, as shown above, suggest that the selective pressure exerted on DF patients’ oral microbiome is relatively low, considering the F level at which fluoride resistant strains are artificially induced. An interesting finding is that higher levels of *Streptococcus*
*mutans* were recorded in the caries-free population of the fluoridated community compared with the non-fluoridated community [96]. The author attributed this phenomenon to the development of inert fluoride-resistant bacteria. Whether fluoride-resistant bacteria pose a threat to oral well-being remains an unsolved problem, with findings on different strains supporting opposite views [97]. To this end, more attempts should be made to substantiate the existence of fluoride-resistant oral bacteria in DF patients.

## 6. Post-Treatment Susceptibility

DF impacts both function and esthetics. The esthetic perception might have measurable psychosocial effects on many patients and negatively affect their quality of life [98]. To cope with this disease, several treatment strategies were proposed, depending on the lesion severity [99]. These treatment measures involve dental bleaching, microabrasion, resin infiltration, composite restorations, veneers, and prosthetic crowns. But such treatments may tip the balance between cariogenic and cariostatic states. For instance, the appliance of restorative materials might alter the surface roughness of dental enamel. A higher roughness will accumulate bacterial plaque more easily. Thus, a proper polishing and finishing procedure is clearly warranted in the dental restorative treatment [100]. Another interesting treatment, microabrasion, starts with the use of etching gels and subsequently applies pumicing with a slow rotation handpiece. As mentioned above, the enamel surface layer might play a more critical role in fluorotic teeth. Removing this layer might cause cariogenic attack to some extent. In fact, a study showed that, after receiving microabrasion treatment, total and ionized Ca, and P concentrations in DF patients’ saliva were significantly raised [101], denoting the demineralization of enamel. This study proved the need for dental practitioners to monitor demineralization and adopt relevant prophylactic measures during therapeutic management, especially among patients with more severe form of DF.

## 7. Conclusions

In this review, we aimed to highlight a more comprehensive strategy in detecting the caries susceptibility of endemic fluorosis populations. A lot of attention was given to the enamel composition and structure, while less focus fell on individual mastication patterns, eating habits, and microorganism factor. Previous studies have intensively centered on whether fluorotic teeth were more resistant to acid, mainly adopting the acid-etching manner, and ignored ambient intraoral conditions and the interplay between cariogenic bacteria and fluorotic enamel (especially how cariogenic bacteria may react to the relatively high fluoride environment and the specific enamel). Thus, research like this may only provide partial evidence for this issue. Intraoral models should be established in futural studies to better simulate real-life cariogenic challenges in order to get a more comprehensive view of this process. As with all these evident or hypothesized protective/pathological factors listed above, the balance between caries progression and reversal in DF patients’ case is more delicate and intricate compared with the non-DF population. The trend that fluorosis with higher scores is more prone to caries due to more severe post-eruptive changes is noticed. Further studies and practice are encouraged to weigh different factors and customize a quantitative analysis in order to get an accurate result of individual caries susceptibility (including the post-treatment condition). Furthermore, given the high-fluoride level in endemic areas, another balance exists for DF patients: between the prevention of caries and skeletal fluorosis, both of which result from excessive fluoride intake. However, unlike its dental counterpart, skeletal fluorosis can be induced throughout life, affecting both children and adults, suggesting a cumulative effect of fluoride. Starting with latent symptoms, at an advanced stage this disease can incur severe outcomes such as various degrees of locomotive disability [11,102]. Bearing this balance in mind, where a higher susceptibility has been confirmed in one patient, systemic fluoride usage should be avoided and replaced with topical supplements such as fluoride dentifrice, which has been proven to be effective in slowing down the demineralization process in vitro [103].

## Figures and Tables

**Figure 1 antibiotics-10-01047-f001:**
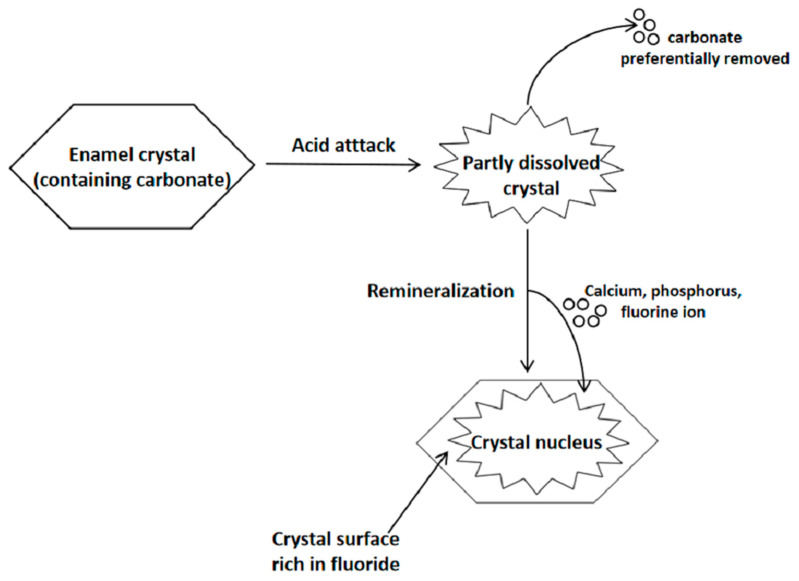
Schematic diagram of the de/remineralization process. The process of remineralization will yield fluoride-rich, carbonate-deficient enamel surfaces.

**Figure 2 antibiotics-10-01047-f002:**
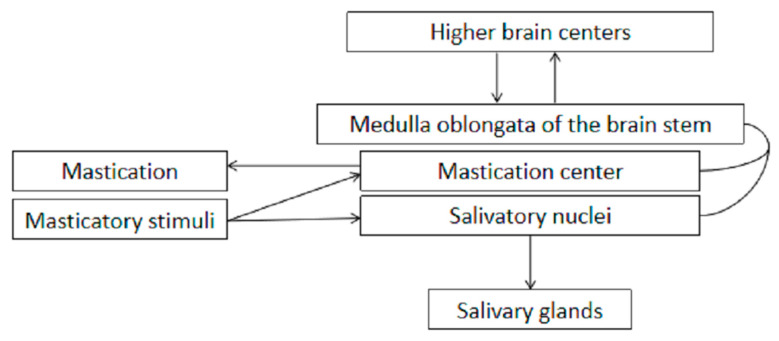
Schematic diagram explaining how mastication influences salivary secretion.

## Data Availability

No new data were created or analyzed in this study. Data sharing is not applicable to this article.

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
