# Peer review of "A Qualitative and Comprehensive Analysis of Caries Susceptibility for Dental Fluorosis Patients"

_antibiotics, 2021, doi:10.3390/antibiotics10091047_

Round 1
Reviewer 1 Report
The review highlighted comprehensively the caries susceptibility of the dental fluorosis patients. The result was interesting as the susceptibility is influenced by many factors of an individual. The work has been well documented to support the aim of the study. The analysis was thorough to explain the effect of each factor in contributing the development of caries in fluorotic patients. Therefore, I recommend acceptance after minor revision.
L37: Introduce the objective and the question of research more clearly.
Methods: Please mention the criteria of selection of the related studies in the analysis of this work.
Author Response
August 25, 2021
Dear editor and reviewers,
Thank you for considering the revised version of our manuscript entitled “A Qualitative and Comprehensive Analysis of Caries Susceptibility for Dental Fluorosis Patients”, by Qianrui Li et al. for publication in Antibiotics. We are grateful to the editor and reviewers for pointing out some important modifications needed in the manuscript. We have thoughtfully taken into account these comments and have made revision marked in red in the paper. Below you will find out point-by-point responses to the reviewers’ comments:
Responses to the reviewers’ comments:
Reviewer #1:
Comment 1:
L37: Introduce the objective and the question of research more clearly.
Response 1:
Thanks very much for your nice suggestion. We have added the question of research and the objective directly on L37.
Comment 2:
Methods: Please mention the criteria of selection of the related studies in the analysis of this work.
Response 2:
Although we have synthesized findings from different approaches to answer a given research question, we mainly consider our study as a narrative and integrative review. Thus, no rigorous selection process was involved. However, we have briefly described our search methods and the basic selection criteria on L51.
Reviewer 2 Report
The subject of this review is interesting, however, the structure is lacking in the organizational department.
English should be checked by a professional.
Please see attached pdf for a detailed suggestions list

Author Response
Comment 1:
The abstract does not contain enough information on the chosen topic.
Response 1:
We have newly given a brief introduction on the main aspects of our analysis.
Comment 2:
Throughout the text “AND” is used excessively and improperly.
Response 2:
Thanks very much. We feel sorry for our mistakes. All the misuse cases of “AND” has been checked and deleted.
Comment 3:
First time using the abbreviation, write what it stands for.
Response 3:
Thank you for the recommendation. The meaning of this abbreviation haven been added at the beginning of the introduction.
Comment 4:
Two places in the introduction are marked as unclear and need to be reformulated.
Response 4:
We apologize for the confusion. The considerations on which we based to determine the focused DF situation have been numbered as a, b, c, to present more clearly. Moreover, we have added more details to describe the function of placental barrier.
Comment 5:
On page 5, line 198 (the original manuscript), “As” should be “a”.
Response 5:
Thanks for your correction. We feel sorry for our mistake. We have revised it.
Comment 6:
Suggestion on further discussing other saliva components that might contribute to dental fluorosis. Suggested literature: Nicolae V, Neamtu B, Picu O, Stefanache MA, Cioranu VS. The Comparative Evaluation of Salivary Biomarkers (Calcium, Phosphate, Salivary pH) in Cario-resistance Versus Cario-activity. Rev. Chim.(Bucharest). 2016 Apr 1;67(4):821-4.
Response 6:
We sincerely thank you for your recommendation. We have checked the literatures carefully, failing to find any valuable papers studying the calcium and phosphate parameters in DF patients’ saliva. But we have found an article (Ref. 55) proving no variation in salivary pH between DF patients and healthy population and we put it directly after the hypothesis about the relationship between mastication and saliva buffer capacity (page 7, line 295) to provide an opposite evidence of our hypothesis. In addition, we have added a study (Ref. 56) finding higher level of secretory immunoglobulin A (sIgA), an important antibacterial component, in DF patients’ saliva. We didn’t cite the suggested article since no research was found to study these parameters (except the salivary pH, which we have already included in our revised manuscript) among DF patients, but we found it useful to explain the role of mucin in saliva and was cited in the part discussing the dynamic changes of saliva with age (Ref. 78, page 8, line 313).
Comment 7:
More focus should be put on differences between saliva in different age gropus, I suggest: Solomon SM, Bataiosu M, Popescu DM, Rauten AM, Gheorghe DN, Petrescu RA, Maftei GA, Maglaviceanu CF. Biochemical Assesment of Salivary Parameters in Young Patients with Dental Lesions. Revista de Chimie. 2019 Nov 1;70(11):4095-7.
Response 7:
We think it’s an excellent suggestion. Given that dental caries is a chronic infectious disease and we previously analyzed the dynamic post-eruptive changes of fluorotic enamel, it’s very meaningful to detect how the quantity and quality of saliva change with age does. We have added this part at the end of the saliva factor chapter. Unfortunately, during our search for relevant literatures, no studies were targeted at this specific population. Therefore, we were forced to study these changes in healthy population and we supposed some similarities might be shared with these two populations. Then we sought to interpret the relationship between these parameters and dental caries. The recommended article was cited (Ref. 75, page 7, line 305), as we found it really helpful to explain the protective role of calcium and phosphate in saliva.
Comment 8:
The authors should add a subchapter detailing restorative materials and dental fluorosis. I suggest citing: Taraboanta I, Stoleriu S, Nica I, Georgescu A, Gamen AC, Maftei GA, Andrian S. Roughness variation of a nanohybrid composite resin submitted to acid and abrasive challenges. International Journal of Medical Dentistry. 2020 June 1;24(2):182-87.
Response 8:
Thank you for pointing this out. We have added the suggested content as a chapter named “post-treatment susceptibility” to the manuscript. In this chapter we mainly focused on treatments and materials for DF that might tip the balance between cariogenic and cariostatic states. The suggested citing has been adopted (Ref. 100, page 9, line 398), because it justified the importance of polishing and finishing restorative materials, otherwise they will accumulate bacteria more easily.
Reviewer 3 Report
The purpose of this qualitative analysis is well described.
The authors have taken into account and analyzed correctly the various factors mentioned and involved on the caries susceptibility of DF topic.
Author Response
Comment:
The purpose of this qualitative analysis is well described. The authors have taken into account and analyzed correctly the various factors mentioned and involved on the caries susceptibility of DF topic.
Response:
Thank you very much for the positive comment.
Round 2
Reviewer 2 Report
The manuscript has been improved